# GIS-Facilitated Effective Propagation Protocols of the Endangered Local Endemic of Crete *Carlina diae* (Rech. f.) Meusel and A. Kástner (Asteraceae): Serving Ex Situ Conservation Needs and Its Future Sustainable Utilization as an Ornamental

**DOI:** 10.3390/plants9111465

**Published:** 2020-10-29

**Authors:** Katerina Grigoriadou, Virginia Sarropoulou, Nikos Krigas, Eleni Maloupa, Georgios Tsoktouridis

**Affiliations:** Laboratory of Protection and Evaluation of Native and Floricultural Species, Balkan Botanic Garden of Kroussia, Institute of Plant Breeding and Genetic Resources, Hellenic Agricultural Organization (HAO)-DEMETER, P.C. 570 01 Thermi, P.O. Box 60458 Thessaloniki, Greece; vsarrop@gmail.com (V.S.); maloupa@bbgk.gr (E.M.); gtsok1@yahoo.co.uk (G.T.)

**Keywords:** rooting of cuttings, seed germination, in vitro propagation, plant growth regulators, geographical information systems, IUCN, Greece

## Abstract

Conservation and sustainable exploitation of threatened endemic plants with medicinal and/or horticultural/ornamental value can be achieved through the development of effective propagation protocols. After unveiling the bioclimatic preferences of *Carlina diae* (Asteraceae) with geographic information systems (GIS), four propagation trials were conducted using seeds of this endangered local Cretan endemic for in vivo and in vitro germination, as well as seasonal vegetative propagation trials (softwood cuttings) and micropropagation (nodal explants). Seed germination was accomplished at a level of 77–90% in vivo (30 days) and 96% in vitro (10 days) using an MS medium with 2.9 μM gibberellic acid (GA_3_). The optimum treatments for cuttings’ rooting were 1000 and 2000 ppm indole-3-butyric acid (IBA) (11–16 roots, 2–3 cm long, 100% rooting) within 40 days in mist. In vitro shoot propagation exhibited a 2.8 proliferation rate after six successive subcultures on an MS medium with 2.9 μM GA_3_. Both ex vitro rooting and acclimatization were successful in 40 days, with 96% microshoot rooting and an equal survival rate. The GIS-facilitated effective species-specific propagation protocols developed in this study can consolidate the perspective of successful re-introduction of ex situ-raised material of *C. diae* into wild habitats and may serve its sustainable exploitation for high-added value ornamental products.

## 1. Introduction

The genus *Carlina* L. (Asteraceae) usually comprises thistle-like plants (annuals, biennials or perennials), often with prickly foliage and daisy-like flowers (compound inflorescences of ligulate and tubular florets) with spiny phyllaries (often leaf-like) that originally are wild growing in Europe, the Mediterranean region and Asia [1,2].

The ornamental value of *C. vulgaris* L. subsp. *vulgaris* (synonym *C. sylvestris* Bubani) and *C. lanata* L. has been reported in Europe as early as the 1620s [3], and *C. acualis* L. subsp. *acaulis* (synonym *C. alpina* Jacq.) was introduced in 1759 as an ornamental of magnificent appearance in gardens of Scotland [4]. Nowadays, plant trade over the internet [5] of some *Carlina* species is extant, mainly as seeds for home gardening or landscaping.

From an ethnobotanical point of view, the flowering heads of *Carlina* spp. are often used globally as a substitute for artichoke and are cooked [6,7,8]. From a medicinal viewpoint, the root of *C. acualis* (and/or *C. acanthifolia* All.), the so-called *Carlinae radix*, has been recognized since ancient and medieval times and is to be found in renaissance botanical books, several pharmacopoeas and contemporary folk medicinal traditions around Europe [9]. *Carlinae radix* has mainly been used as a diuretic, diaphoretic, stomachic and externally for the treatment of skin inflammations, as well as against toothache, for treating cholecystopathy and gastrointestinal disturbances, while the root’s essential oil is proved to have antimicrobial, antibacterial, anti-inflammatory, anti-ulcer, and antioxidant activities [9,10].

Due to protection status or limited extent of occurrence in several countries and to address the commercial interest for *Carlina* spp. (both medicinal and ornamental), various propagation and cultivation systems (field cultivation, hydroponics, in vitro cultures) have been reported to date [11]. These mainly include micropropagation of the stemless *C. acaulis* subsp. *acaulis* using immature zygotic embryos [12]; microplant production of *C. acaulis* subsp. *acaulis* and *C. acanthifolia* subsp. *utzka* (Hacq.) Meusel and A.Kástner) from callus cultures using root and stem explants [13]; in vitro regeneration of *C. acaulis* subsp. *caulescens* (Lam.) Schübl. & G.Martens using shoot tips, fragments of hypocotyls, cotyledons and roots from sterile seedlings [14,15,16]; shoot multiplication [17] and rooting [18] of *C. acanthifolia* subsp. *utzka* using shoot tip and hypocotyl explants. At the same line, sexual propagation of *C. vulgaris* [19] and *C. acaulis* [11] has succeeded, and seed dormancy was studied as well [20]. The germination of *Carlina* seeds is generally reported to be very high, i.e., 80–100% for *C. corymbosa* L. [21,22], 94% for *C. acaulis* subsp. *acaulis* [22] and 100% for *C. acanthifolia* [22,23], *C. vulgaris* [22,24], *C. biebersteinii* Bernh. ex Hornem., *C. involucrata* Poir., and *Chamaeleon macrocephalus* (Moris) Sch. Bip. subsp. *macrocephalus* (synonym *Carlina macrocephala* Moris) [22].

Apart from the above-mentioned widespread *Carlina* taxa, this genus includes another 20 single-country endemics (69% of the total taxa in the genus) and therefore uncommon, range-restricted species and subspecies [25,26], such as: *C. biebersteinii* subsp. *sudetica* Kovanda confined to parts of the Czech Republic [22]; *C. acanthifolia* nothosubsp. *lecoqii* (Arènes) B. Bock confined to parts of France; *C. canariensis* Pit., *C. texedae* Marrero Rodr., *C. falcata* Svent. and *C. xeranthemoides* L. f. confined to the Canary Islands in Spain (Gran Canaria (the first two), La Palma and/or Tenerife, respectively)); *C. salicifolia* (L. f.) Cav. confined to the Canary Islands (Spain) and Madeira (Portugal); *C. frigida* Boiss. and Heldr. subsp. *fiumensis* (Simonk.) Meusel and A. Kástner confined to parts of Croatia; *C.* × *szaferi* Jasiewicz and Pawl. confined to parts of Poland; *C. guittonneaui* Dobignard confined to parts of Morocco; *C. kurdica* Meusel and A. Kástner confined to north-west Iraq; *C. nebrodensis* Guss. (unplaced name according to Plants of the World Online) and *C. sicula* Ten. subsp. *sicula* confined to Sicily, Italy; *C. oligocephala* Boiss. and Kotschy subsp. *oligocephala* and *C. oligocephala* subsp. *pallescens* (Wettst.) Meusel and A. Kástner confined to Anatolia, Turkey; *C. pygmaea* Holmboe confined to parts of Cyprus; *C. barnebyana* B.L. Burtt and P.H. Davis, *C. curetum* Heldr. subsp. *curetum, C. diae* (Rech. f.) Meusel and A. Kástner and *C. sitiensis* Rech. f. confined to islands of Greece (Crete and small islets and/or Karpathos and/or Kasos islands). Among the latter, *C. diae* is a rock-dwelling plant of inaccessible cliff faces and steeply sloping calcareous rocks close to the sea (0–150 m) which is confined exclusively to the islets of Dia, Dragonada and Gianisada off the north coast of Crete (Figure 1a,b), with few isolated individuals also found at Cape Movros and in a gorge close to Toplou Monastery over Karoumpes bay in north-eastern Crete [27].

The selection of *C. diae* as a focal species for this study is primarily due to the considerable scientific value associated with its uniqueness. In the first place, it is a very rare, local (Cretan) endemic plant with a total population size <1000 mature individuals remaining in the wild (severely fragmented subpopulations, each with <250 mature individuals) [27]. All its subpopulations are included in the NATURA 2000 sites GR4310003 and GR4320006 and it is protected by the Bern Convention (Appendix I) and the Greek Presidential Decree 67/1981. However, *C. diae* is threatened with extinction due to overgrazing and it is assessed as endangered with decreasing population trend [27]. Secondly, it is considered as a Tertiary relict of the small and ancestral subgenus *Lyrolepis* Meusel and Kästner., which represents “an ancient type compared to (the rest of) *Carlina* spp.” [28], thus rendering it as de facto “unusual” and therefore unique and attractive among other similar plants. It is not cultivated as a rare ornamental plant [5] and there are only a couple of specimens conserved in ex situ facilities.

Therefore, there is need for its effective ex situ conservation [29,30,31,32]. In this context, the present study investigates the sexual and asexual reproduction of *C. diae,* a potential medicinal plant with conservation priority and highly promising ornamental value, using geographical information systems (GIS) to unveil its seasonal bioclimatic preferences. This aim to present a consolidated multiplication process serving ex situ conservation efforts and facilitating sustainable exploitation needs.

## 2. Results

### 2.1. Seasonal Bioclimatic Preferences of C. diae

The GIS-derived bioclimatic profiling of *C. diae* indicated that during winter (coldest quarter: December, January and February), the populations of this species experience in the wild habitats 233 mm of mean precipitation (high water availability) and monthly temperature minima from 5.6 to 8.2 °C (see Appendix A), representing its natural lowest temperature limit. Similarly, during spring (driest quarter: March, April and May), *C. diae* is naturally adapted to 5.7 mm of mean precipitation (suggesting limited water requirements and natural adaptation to dry arid conditions) and monthly temperature means increasing gradually from 10.9 °C (March) to 15.0 °C (April) and 18.0 °C (May), indicating a period of active growth probably till early summer (Tmean of June: 22.2 °C) (Appendix A). During summer (warmest quarter: June, July and August) it is adapted to 2 mm of mean precipitation (natural lowest water requirements) and mean monthly temperature maxima from 26.0 to 28.3 °C, representing its natural highest temperature limits (see Appendix A). During autumn (wettest quarter: September, October, November), *C. diae* experiences 244 mm of mean precipitation (highest water availability) combined with monthly temperature means decreasing from 21.7 to 14.1 °C, thus probably suggesting the progressive entrance into a “dormancy”-period of restricted (non-active) growth during the winter months with monthly temperature means decreasing further below 10 °C (see Appendix A).

### 2.2. In Vivo Seed Germination

The germination of seeds of *C. diae* in the wild habitats most probably occurs naturally during late winter or early spring, a period with high water availability and mean temperatures ranging from close or above 10 °C up to 18.0 °C (see Appendix A). With annual mean temperature and temperature annual range at around 18–20 °C, the GIS-derived bioclimatic profile of *C. diae* indicated the selection of appropriate temperatures for in vivo seed germination testing (19 ± 2 °C).

The seed germination of *C. diae* was completed within 30 days, reaching the highest percentage in less than 15 days at 19 ± 2 °C and RH 80–90%. The initiation of germination (visible signs of sprouting) in both treatments was observed on the 6th day. The germination rates for the 15th and 30th day were significantly higher (70.00–90.00%) after the drying and cold storage of seeds (4–5 °C, RH < 5%) compared to those of direct sowing after collection from the natural environment and drying (63.33–77.33%). The highest germination rate (90.00%) was observed in the treatment of cold storage for 80 days. The duration of germination and the cold storage had a significant effect on the number of germinated seeds and germination rates (%) (*p* ≤ 0.001), whereas their interaction did not show any substantial change (*p* = 0.997 > 0.05) (Table 1, Figure 2a–c).

### 2.3. In Vitro Seed Germination and Microshoot Development

Infections appeared after 10 days of culture. Only one out of 29 test tubes containing GA_3_-free medium and five out of 50 test tubes with 750 μM GA_3_ were infected with fungi and bacteria, thus the disinfection procedure of seeds was successful at a level of 90–98% in both media tested. The in vitro germination process was initiated on day 4 after establishment in both culture media. The optimum germination rate for seeds cultured in MS medium [33] with GA_3_ was 96.55% after 10 days, while in GA_3_-free medium, it was 100% (not statistically significant) after 30 days of culture, indicating that GA_3_ caused a significant acceleration of germination and simultaneously much earlier radicle protrusion (Table 2, Figure 2d,e).

Nodal explants (*n* = 40) cultured in MS medium with 0.9 μM BA + 2.9 μM GA_3_ suffered severe hyperhydricity problems, resulting in 20 non-vitrified explants after a 4-week culture period. The vitrified cultures were maintained in culture for five subsequent sub-cultures, showing browning/necrosis symptoms and disintegration. On the other hand, the 25 initial explants cultured in MS medium with only 2.9 μM GA_3_ (BA-free) responded better (less intense hyperhydricity symptoms), yielding progressively 13, 14, 22, 50, 60 and 70 non-vitrified explants after six successive subcultures (4 weeks/each). Therefore, within a 6-month period, the initial number of stock mother plants was raised from 13 to 70 (>5-fold increase) in a 2.9 μM GA_3_ MS medium (Figure 2f).

Both ex vitro rooting and acclimatization were successful in 40 days, with 96% microshoot rooting and an equal survival rate. Two months later, the average height of plants was approximately 10 cm and they were ready to be transplanted in 1 L pots for further development.

### 2.4. Vegetative Propagation by Cuttings

Compared to the control, the rooting of the *C. diae* cuttings was optimal (100%) using 1000 ppm IBA and decreased (71.43% and 57.14%) when 2000 or 4000 ppm IBA was applied, respectively. All IBA treated cuttings produced a 2–3-fold increase in the number of roots per cutting (11.43–17.00) compared to the control (5.00 roots/cutting). Root length was adversely influenced by the IBA application (2.08–2.97 cm) compared to the control (4.26 cm). The treatment with IBA, regardless of concentration, caused a 1.5–2 cm decrease in the length of roots compared to the untreated cuttings. No symptoms of browning and necrosis were observed in the control and treatments with 1000 or 2000 ppm, while 42.86% necrosis was observed when cuttings were treated with 4000 ppm IBA. Taking all macroscopic parameters into account, the 2000 ppm IBA treatment (16.00 roots; each 2.08 cm long; 100% rooting) proved to be the optimum concentration for the rooting of cuttings after 40 days during late spring-early summer (Table 3, Figure 3).

### 2.5. Ex Situ Conservation and GIS-Derived Data

After the trials, the propagated plants continued developing and acclimatized without problems in a semi-shady (50%) place at the nursery of the BBGK (sea level) during spring and summer and throughout the year. The propagated plants withstood the increased summer temperatures at the sea-level nursery of the BBGK well, which were within the almost compatible range (Tmin: 15.7–17.9 °C, Tmax: 28–30.7 °C) with regard to those of the natural habitat of *C. diae* (see Appendix A). The propagated plants representing two genetic stocks (one derived through vegetative propagation and another produced through the seed germination trials) were successfully acclimatized in local conditions of northern Greece. All plants developed flowers that were freely pollinated and produced seeds from the first year of their ex situ cultivation, allowing new seed collection for long-term storage.

## 3. Discussion

The development of propagation protocols for rare and threatened species is imperative for effective ex situ conservation [34,35,36,37]. The current study focused on the GIS-facilitated propagation of the rare and threatened (endangered) local Cretan endemic *C. diae* by seeds in in vivo and in vitro conditions, as well as by cuttings. The results presented herein demonstrate the development of efficient propagation protocols regarding *C. diae*, serving either conservation purposes or its sustainable exploitation as a new flower crop in the horticultural/ornamental industry and as a potential medicinal plant for the pharmaceutical sector. Both vegetative and sexual reproduction methods were successfully applied in *C. diae* and the optimum conditions for both systems have been determined concisely. It is suggested that consideration must be given to the environment-specific needs of the plant species under propagation and conservation study [38]. In the absence of published species-specific studies, the GIS application used in our study has offered the potential to unveil the ecological preferences of *C. diae* in a meaningful way for ex situ conservation and horticulture. This helped to improve the common routine exercised in botanical gardens when dealing with new species (trial-and-error losses of plant material). In the case studied here, valuable and rare plant material was used which was acquired only in limited quantities in order to minimize possible risks for the endangered wild population. The GIS application used has contributed to the selection of: (a) appropriate temperatures for greenhouse seed germination, rooting of cuttings and cultivation, (b) appropriate period for acclimatization and transplanting of plantlets produced in vitro, and (c) spatial positioning within the nurseries and the ex situ conservation facilities of the BBGK. Our study illustrates how GIS may facilitate the seed germination, rooting of cuttings and the in vitro propagation of limited plant material of conservation priority species (such as *C. diae*), and in addition provides ecologically based guidelines for its ex situ cultivation.

The seeds of *C. diae* do not appear to have dormancy restrictions, as they germinated at a satisfactory rate under various conditions examined (both in vivo and in vitro) (Table 1 and Table 2). Although the germination capacity of *C. diae* seeds was higher than 90% in both in vivo and in vitro systems, the cold storage of *C. diae* seeds for 80 days enhanced the germination rate in vivo (Table 1) and the use of GA_3_ accelerated the in vitro germination process (Table 2). This is quite common for some species in the Asteraceae family, where the efficiency of seed germination and seedling growth is low because they are highly dependent on various biological and environmental factors [39]; therefore, treatments with plant growth regulators (PGRs) such as GA_3_ promote their germination [40]. The maximum in vivo germination rates of *C. diae* seeds were achieved 30 days after sowing for both trials, although the experiments were evaluated for two months. Therefore, no sign of dormancy was observed for *C. diae*, unlike many other native species in the Asteraceae family, which need special treatments to break the seed dormancy [41]. Our results are in accordance with those of Fournaraki (2010) [42], in which *C. diae* seeds germinated well under laboratory conditions in all temperatures tested (10, 15 and 20 °C) in white light/dark (12 h/12 h) and at constant darkness (24 h), with germination delayed at the lowest temperature (10 °C) and all germination rates high (100%) at all temperatures. The species’ seeds are not characterized by primary normal dormancy and exhibit orthodox storage behavior [42]. In line with that, the germination data in Petri dishes containing 1% agar of nine *Carlina* species [22] show high germination success (80–100%) in different photoperiod regimes (8 h light/16 h darkness or 12 h light/12 h or 24 h darkness) within a short period of 7 to 42 days and within a wide range of temperatures (5–25 °C).

This study describes for the first time a complete production process regarding *C. diae* in vitro plants, from seed germination to acclimatization. Disinfection of *C. diae* seeds provided a high percentage (90–98%) of non-infected seeds which were used as initial plant material for micropropagation. Although contamination is a limiting factor in general for micropropagation [43], the disinfection of seeds used as explants in different *Carlina* spp. is a common process [11,15,16]. The initiation of the in vitro germination was accelerated, and it was achieved as early as on day 4 using an MS medium with GA_3_ with maximum germination of 96% after 10 days, in contrast to 100% germination in a 30-day period on GA-free medium. The results are similar to those regarding *C. acaulis* where the tested achenes with viable seeds germinate very well (94%) on MS medium supplemented with 2.9 μM GA_3_ after 10 days of culture in continuous light (26 ± 1 °C) [15] as well as to those concerning *C. acanthifolia* subsp. *utzka* [11,17].

Some phenotypic effects produced by light in plants are induced by PGRs and especially gibberellins (GAs). It is reported that processes such as germination, de-etiolation, root/tuber formation or stem growth are complex outcomes of interactions between light and gibberellins [44]. Compared to the in vitro seedlings produced on plain MS in our study, the in vitro seedlings of *C. diae* derived from an MS medium supplemented with 750 μM GA_3_ showed a healthier appearance, larger internodes and generally more robust and vivid habit and, therefore, a better quality (Figure 2d,e). At the same line, it is reported that *C. acaulis* plantlets grown under a 16h photoperiod and treated with 100 μΜ GA_3_ may develop maximum stem length increase due to the increased number and length of internodes [12]. Such a trend may be attributed to GA_3_, which is involved in stem elongation, triggered by increased photoperiod and thus a shortening of darkness [45].

In the proliferation phase of *C. diae*, the use of BA in combination with GA_3_ resulted in hyperhydricity and complete destruction of cultures of *C. diae* after five subcultures compared to the MS medium with GA_3_ alone, which allowed a satisfactory rate of microshoot production. This trend was also observed during shoot proliferation of *C. acanthifolia* subsp. *utzka* [17]. Moreover, inhibition of shoot elongation by BA has been also reported in *C. acaulis* [15,16] and other Asteraceae species [46,47]. However, the use of BA has been reported in other species of Asteraceae [48,49,50], as it may overcome apical dominance, promoting axillary bud development [12,51,52]. In vitro culture of *C. acaulis* subsp. *caulescens* may be achieved with a satisfactory bud multiplication in an MS medium supplemented with 4.44 μM BA and 1.14 μΜ IAA [12]. The best morphogenetic response of *C. acaulis* can be observed when shoot tips and other explants are cultured in MS containing 13.32 μM BA and 0.54 μM NAA producing a 6.1 multiplication rate during the first five subcultures [16]. In the current study, the *C. diae* nodal explants cultured in MS medium with 2.9 μM GA_3_ (BA-free) generated the highest shoot proliferation rate, achieving a 5.38-fold increase in a six-month period.

Exogenous auxins are commonly used to improve natural rooting efficiency of stem cuttings, but it has been demonstrated in various plant species that relatively high auxin concentrations are required only during the induction phase, while PGRs during development have a rather negative effect [53]. In this study, the positive response of *C. diae* cuttings (an increase in rooting rate from 70 to 100%) with the application of 1000 or 2000 ppm IBA could be due to the low supplement of endogenous auxins in the shoots of the plant and to the fact that auxin might have interacted positively with the application of exogenous rooting hormone [54]. Similarly, the promoting effect of IBA (500 ppm) on the rooting ability of another species of the Asteraceae (i.e., *Stevia rebaudiana* (Bertoni) Bertoni) has been reported [55]. Such a trend could either be due to the positive effect on root initiation, the formation of more and uniform roots [54] or due to acceleration of nutrient translocation from the upper part of the cuttings to their basal ends by increasing the activity of energy-producing enzymes [56].

A previous study related to the effect of IBA and NAA on rooting of *Chrysanthemum × morifolium* (Ramat.) Hemsl. (also, Asteraceae) terminal cuttings showed that cuttings treated with 400 ppm IBA are associated with higher rooting percentage [57]. IBA is the most extensively used auxin to enhance root induction in cuttings due to its high ability to stimulate rhizogenesis, weak toxicity and great stability with respect to NAA [58]. The percentage of cuttings with root formation of *C. diae* after application with either 1000 or 2000 ppm IBA was substantially higher than that obtained in the untreated cuttings, demonstrating the importance of synthetic auxin for asexually produced *C. diae* plants. However, in our study the highest applied IBA concentration of 4000 ppm negatively affected rooting by decreasing the percentage and many side-effects were presented. It is reported that IBA might be toxic to certain softwood cuttings taken from perennial plant species, resulting in poor or absent growth and/or mortality of the cuttings due to an antagonism between the high concentration of exogenous IBA and the endogenous auxin of the plant [59]. A similar explanation suggests that high concentrations of exogenous auxin levels in the cuttings might disturb the hormonal metabolism, exerting an inhibitory effect on root formation [60].

In the present study, the maximum number of roots per rooted cutting (16–17) was recorded under 2000 and 4000 ppm IBA. This finding suggests that the treated cuttings with auxin at appropriate concentrations induced early and better root initiation (i.e., more roots per cutting) [53]. The application of IBA may indirectly favor rooting by raising the translocation speed and sugar movement from the apex towards the base of the cutting [61]. The good rooting ability of basal cuttings could be due to higher sugar reserves or due to the accumulation of natural auxins in the shoot bases or other segments promoting relatively low levels of rooting inhibitors [62,63], or may be due to juvenility factors [64] found along the plant’s stem [65]. The positive effect of IBA was also confirmed in the case of ex vitro rooting of *C. diae* microshoots. This treatment allowed faster production of plants for *C. diae*, by omitting the rooting stage in vitro and offered benefits regarding the hardening of microplants during the rooting process in the mist chamber in vivo. Therefore, this technique may be also recommended for the mass propagation of other similar perennial native plants.

With efficient ex situ conservation of *C. diae* and successful propagation protocols that can be applied at a commercial scale, its sustainable exploitation may be addressed. *C. diae* has noteworthy ornamental features which are usually appreciated by the horticultural industry: the wild-growing individuals are densely white- or silver-felted dwarf and rigid shrubs (strongly wooded below), with numerous, short, non-flowering densely crowded branches, and sparsely leafy, erect, flowering stems 40–60 cm high that are sparingly branched above. Each flowering stem has a small flat-topped cluster of one to four flower heads (Figure 1d) of tubular dull yellow disc florets surrounded by several rows of bracts (the outer small, lanceolate, entire or with a few small lobes; the inner bright yellow, shining, scarious, and radiating) [27,66]. Flowering from June to August, it has potential as an attractive rock garden plant blooming throughout summertime [66]. It may also prove to be suitable for xeriscaping due to its obligatory rock-dwelling habit and concomitant limited needs for nutrients and water, thus rendering it advantageous for gardening in arid areas with low rainfall and scarcity of water (see Appendix A). The proximity of its wild populations to sea level (coastal chasmophyte, though not halophyte) suggests that moderate salt spraying may be tolerated, thus rendering *C. diae* as suitable for gardening in coastal areas. Moreover, it could possibly be introduced as an interior pot plant, being able to withstand the low humidity indoors during winter. Furthermore, there are reports of free hybridization with *C. vulgaris* in old successful glasshouse cultivation in Germany (a completely different thistle-like plant) [1], and thus shows promise for the breeding of new ornamentals.

## 4. Materials and Methods

### 4.1. Botanical Collections

The botanical collection was performed using a special permit of the Balkan Botanic Garden of Kroussia (BBGK), which is issued yearly by the Greek Ministry of Environment and Energy. Seeds and fresh soft wood cuttings of *C. diae* were collected in late August of 2018 from wild-growing populations found as rock-dwellers in the so-called “middle bay”(Paralia Panagias; 35°26′21″ N, 25°13′24″ E) of the islet of Dia (northern Crete) (Figure 1a,b). The collected material (seeds and cuttings) was transferred to the facilities of the BBGK in Thessaloniki (Thermi) and received an International Plant Exchange Network (IPEN) accession number (GR-1-BBGK-19006). To desiccate, the seeds (350 seeds in total, 1.725 g) were maintained for 80 days in a dark chamber at 15 °C and relative humidity (RH) of 15%.

### 4.2. Unveiling the Ecological Preferences of C. diae with GIS

The GIS application developed by Krigas et al. [35,36] was used to reveal the basic ecological preferences of *C. diae* in order to facilitate its ex situ conservation. The exact geographical coordinates (recorded in situ with handheld global positioning system trackers) of the original collection site of *C. diae* (from which the initial propagation material was collected) were imported into GIS and were linked accordingly with bioclimatic data extracted from Lazarina et al. (2019) [67]. This link furnished bioclimatic information regarding the original collection site of *C. diae* (see Appendix A).

### 4.3. In Vivo Seed Germination

The GIS-derived ecological profile of *C. diae* was used to select appropriate seasonal temperatures for seed germination testing. The total number of seeds of *C. diae* was divided into three portions (100 seeds were stored for long-term ex situ conservation and in vitro experiments). The first portion (*n* = 100 seeds, 0.495 g) was used for germination immediately after drying during late autumn (November 2018), while the second one (*n* = 150 seeds, 0.775 g) was kept in a cold chamber (4–5 °C, RH < 5%) for 80 days in order to be used for the second experiment during late winter (February 2019). In both cases, the seeds were saturated in de-ionized water (dH_2_O) for 24h and then sown (4–5 mm depth) in plastic trays containing a substrate of peat (Terrahum, Klassman):perlite (1:1 *v*/*v*) and covered with a layer of vermiculite (1–3 mm). The trays were then placed in a heated bench mist chamber (19 ± 2 °C, RH 80–90%), close to the natural temperatures of spring months, the annual mean temperature and the temperature annual range of the wild habitats of *C. diae* (see Appendix A). Seed germination was assessed every 15 days for two months by calculating the number of seeds with visible germ growth (sprouting). After germination and growth for 60 days, the seedlings were transferred in multiple position seedling trays, containing the same substrate; four weeks later, they were transplanted into larger pots (0.33 L) containing a mixture of enriched peat moss (TS2, Klassman): perlite (3:1 *v*/*v*) and two months later were transferred in 1 L pots with the same substrate.

### 4.4. In Vitro Seed Germination and Microshoot Production

The germination of *C. diae* seeds was also investigated under in vitro conditions. The seeds from the original collection were maintained in a cold chamber (4–5 °C, RH < 5%) for six months. The seeds were saturated in dH_2_O for 24 h at room temperature in the dark and then were disinfected via the following steps: firstly, in a fungicide solution (0.1 g Signum 26.7/6.7 WG, BASF/100 mL sterile dH_2_O) for 30 min; secondly, by immersion in a 70% ethanol solution for 1 min; lastly, in 5% NaOCl for 5 min with agitation and finally rinsed four times with sterile dH_2_O in a laminar flow hood. The basal nutrient culture medium used was the MS (Duchefa Biochemie) [33] supplemented with either 750 μM GA_3_ (Duchefa Biochemie) (*n* = 29 seeds) or without GA_3_ (*n* = 50 seeds). The seeds were placed in glass borosilicate flat-base tubes (25 mm width × 100 mm height) containing approximately 10 mL of the above media and were incubated in a walk-in growth room at 22 ± 2 °C and with a 16h photoperiod (40 μmol m^−2^ s^−1^), emulating in a way the natural spring conditions of the wild habitats of *C. diae* (see Appendix A). Seed germination was assessed on day 4, 10, 20, 30 and 40. The seed cultures were maintained for three months; the obtained seedlings were dissected in shoot nodal segments and transferred to an MS medium supplemented with: (a) 0.9 μM 6-benzylaminopurine (BA, Duchefa Biochemie) and 2.9 μM GA_3_ (40 explants) and (b) 2.9 μM GA_3_ (25 explants) for enhancing shoot proliferation and elongation of explants. The proliferated microshoots were dissected and sub-cultured every four weeks in the same media; shoot proliferation rate was recorded each time, for a period of six months. All the above culture media were enriched with 20 g/L sucrose (Duchefa Biochemie) and the pH was adjusted to 5.8, prior to the addition of 6 g/L agar (Plant Agar, Duchefa Biochemie) as a gelling agent.

Furthermore, the possibility of ex vitro rooting was investigated. The microshoots were rinsed with tap water, treated with 0.2% indole-3-butyric acid (IBA) (commercial formula Radicin, in powder, Fytorgan SA, Greece) and planted in multi-position propagation trays filled with a peat (Klasmann, KTS 1)-perlite 1:1 (*v*/*v*) mixture. The trays were placed in a greenhouse under a 90% RH fog system and 50% shading for 20 days. For the next 10 days, RH was reduced (5%/day) to standard greenhouse conditions (40–50% RH) while light intensity was gradually increased [68,69]. The percentage of successfully acclimatized plants was determined 40 days later.

### 4.5. Vegetative Propagation by Cuttings

The three initial cuttings collected from the wild were treated with 0.2% IBA and they were placed for a 7-week period on a heated bench under mist at 19 ± 2 °C and RH 80–90% (close to the natural temperatures of spring months of the wild habitats of *C. diae,* see Appendix A). Only one out of them formed roots and it was transplanted into a larger pot (0.33 L) containing a mixture of a more enriched peat moss (TS2, Klasmann):perlite mixture at a 3:1 *v*/*v* ratio. After four months, the plant was transplanted into a larger pot (2.5 L), and five new cuttings were derived, and they were rooted with the same procedure. Following this process, adequate stock material (mother plants) was created for further experimentation, genetically identical to the original wild growing plant.

During late spring (late May–June, when active growth occurs in natural habitats) root formation was tested using four different concentrations (0, 1000, 2000 and 4000 ppm) of IBA (Duchefa Biochemie). Softwood tip cuttings 3–4 cm long were immersed for 10 sec in the above IBA solutions (dissolved in 50% ethanol) and were placed in multi-position propagation trays using a 3:1 *v*/*v* peat moss (TS1, Klasmann): perlite substrate. To emulate the spring and summer conditions of the wild habitats of *C. diae* (see Appendix A), the trays were moved on a heated bench (soil temperature 18–21 °C) in a mist chamber (air temperature 18–30 °C, depending on local weather conditions and RH 70–85%) of the BBGK’s greenhouse. The number of roots per cutting and root length were measured after 40 days.

In addition, the seasonal variation in the rooting ability of the cuttings was studied using the auxin IBA at 2000 ppm concentration. Softwood tip cuttings 3–3.5 cm long were excised from mother stock plants maintained in a non-heated greenhouse during three different seasons: early summer (June 2019), autumn (late October/November 2019) and winter (January 2020). In an attempt to imitate as best as possible the natural temperatures prevailing in the wild habitat of *C. diae* throughout the year, the rooted cuttings of *C. diae* were maintained in the greenhouse with annual average temperatures ranging from Tmin > 8 °C to Tmax < 35 °C, which fall within the range of extreme natural temperatures (max temperature of the warmest month, min temperature of the coldest month) that *C. diae* populations experience in the wild (see Appendix A). The same procedure as described previously was followed, but soil and air temperature (T) and RH were different due to seasonal variability of the BBGK’s greenhouse, i.e., during summer: soil T: 18–22 °C, air T: 25–35 °C (close to mean temperature of the warmest and driest quarters and max temperature of the warmest month, see Appendix A) and RH 60–70% (corresponding to the comparatively lower precipitation of the warmest and driest quarters and precipitation of the driest month, see Appendix A); during autumn: soil T: 18–20 °C, air T: 15–25 °C (close to mean temperature of the wettest quarter, see Appendix A) and RH 70–85% (corresponding to the intermediate precipitation of the wettest quarter, see Appendix A); during winter: soil T: 17–19 °C, air T: 5–15 °C (close to mean temperature of the coldest quarter and min temperature of the coldest month, see Appendix A) and RH 85–99% (corresponding to the comparatively higher precipitation of the coldest quarter, see Appendix A). These ex situ seasonal conditions were compromised due to extant technical specifications of the BBGK’s greenhouse and were selected to be as close as possible to the seasonal variation of natural conditions of the wild habitats of *C. diae* (see Appendix A). The number of roots per cutting and root length were measured after 30 days for autumn and summer trials and after 50 days for the winter ones.

Rooted plants from all the above treatments were transplanted into larger pots of 0.33 L and subsequently into 2.5 L containing a mixture of peat (Klasmann, TS2) and perlite (3:1 *v*/*v*) to continue growing. The excess plant material created was planted at the grounds of the Balkan Botanic of Kroussia at sea level in Thermi, Thessaloniki for long-term ex situ conservation purposes.

### 4.6. Statistical Analysis

All experiments were conducted in a completely randomized design and were repeated twice. The reported data are the means of the two experiments. The means were subjected to analysis of variance (ANOVA) using the statistical package SPSS 17.0 (SPSS Inc, Chicago, IL, USA) and compared by using Duncan’s multiple range test. In the in vivo germination trials (a 2 × 2 factorial experiment), the main effect of factors (number of days from sowing and seed pre-treatment/sowing season) and their interaction were determined by the general linear model (two-way ANOVA). In the in vitro germination trials (a 2 × 2 factorial experiment), the main effect of factors (the number of cultures in the MS medium and the presence or not of GA_3_) and their interaction were determined by the general linear model (two-way ANOVA). The vegetative propagation experiments for rooting of the softwood cuttings were consisted of four and three treatments, where each value was the mean of 21 replicates (three groups of seven repetitions).

## 5. Conclusions

Extremely rare single-island or single-mountain local endemics of Asteraceae [69,70], respectively) are threatened with extinction across the world, and in situ conservation coupled with ex situ effective propagation methods are the only apparent avenues to preserve them effectively [69,70]. Given the high seed germination success achieved (both in vivo and in vitro, shortened process due to PGR application) and the effective macro- and micropropagation protocols currently developed, the GIS-facilitated ex situ conservation efforts described in this study for *C. diae* can provide a consolidated reproduction system for this rare and endangered local endemic species of Crete (Greece). The knowledge developed herein (starting with reduced plant material and limited background information) may be further exploited for more applied conservation research focused on future reinforcement of wild populations and/or possible reintroductions in its original habitats. Targeted botanical collections can be designed to acquire selected plant material from different subpopulations of *C. diae* for directed propagation aimed at increased genetic diversity (α-diversity). If necessary, specific neopopulations can be designed accordingly to meet the needs of defined conservation targets or actions. For these, the new plant material needed can be raised in due time under ex situ conditions based on the species-specific propagation protocols described in this study, and the concomitant genetic variability from germination of different seed sets can be exploited accordingly for conservation purposes. In addition, the number of identical individuals per selected genotype can be increased adequately through vegetative propagation protocols as described herein. Apart from the above mentioned outcomes, the results of this study offer also an important stepping stone that may facilitate the sustainable utilization of *C. diae* in the commercial horticulture sector as a new ornamental plant for high added value products related to the gardening of unique plants and landscaping with native plants in arid areas.

## Figures and Tables

**Figure 1 plants-09-01465-f001:**
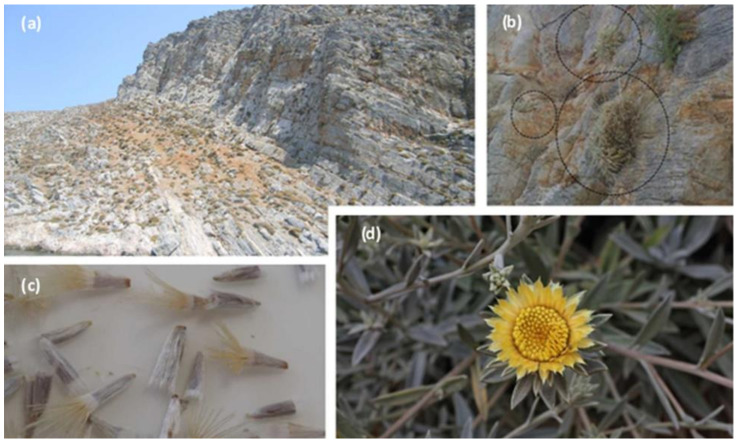
The original habitat of rock-dwelling *Carlina diae* on the islet of Dia, off the north central coast of Crete: (**a**,**b**) Rocky cliff faces and inaccessible steeply sloping calcareous rocks close to the sea (individuals in dashed circles); (**c**) mature seeds (achenes); (**d**) inflorescence (photos (**a**,**b**) by M. Avramakis).

**Figure 2 plants-09-01465-f002:**
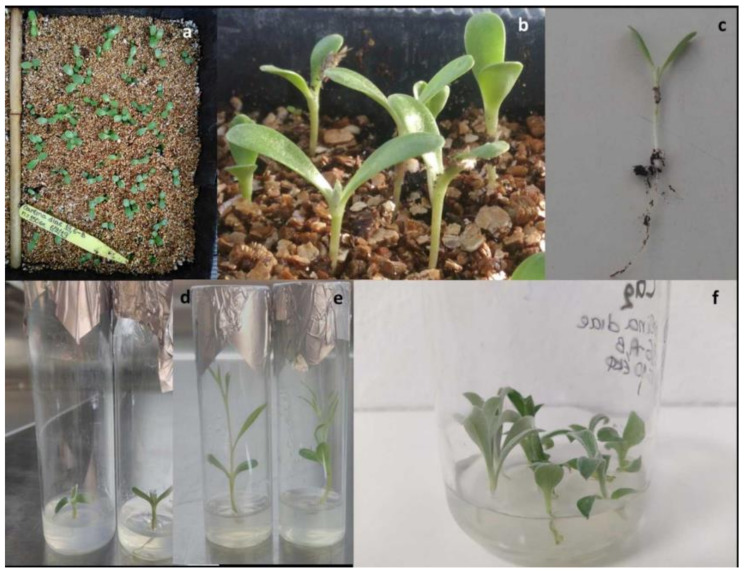
Seed germination of *Carlina diae:* (**a**) in vivo after 20 days; (**b**) in vivo after 40 days; (**c**) in heated bench mist system; (**d**) in vitro in MS culture medium after 40 days of culture without plant growth regulators (PGRs); (**e**) in vitro in MS supplemented with 750 μM gibberellic acid (GA_3_); (**f**) in vitro proliferation phase in MS + 2.9 μM GA_3_.

**Figure 3 plants-09-01465-f003:**
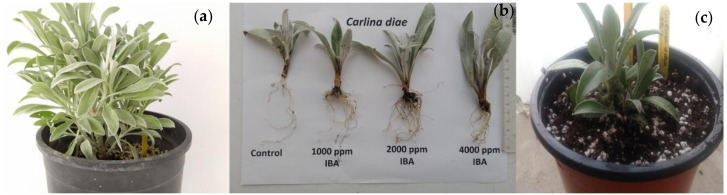
Vegetative propagation of *Carlina diae* softwood tip cuttings: (**a**) Mother plant obtained from rooted initial cuttings of the botanical expedition on the islet of Dia, Crete, Greece; (**b**) effect of IBA concentrations (control, 1000, 2000 and 4000 ppm) on rooting after 40 days of maintenance under mist; (**c**) rooted plants transplanted in 2.5 L after 90 days. The rooting response of cuttings during winter was low, with only 52.38% root formation (approximately a 50% decrease) in a 50-day period. On the contrary, during summer and autumn (when the temperature in the greenhouse was almost constantly above 20 °C), the rooting percentage was similar (100%) within the same 30-day period, but there were differences between the number of roots per cutting and their length. More specifically, the number of roots was higher in summer (13.17) compared to the other two seasons (9.73–9.86) and the root length was longest (3.01 cm) during autumn and shortest during winter (1.37 cm) (Table 4).

**Table 1 plants-09-01465-t001:** Germination of *Carlina diae* seeds collected from the wild after drying (*n* = 90) or after drying and storage for 80 days in a cold chamber (4–5 °C, relative humidity (RH) < 5%, pre-treatment in dH_2_O for 24h) (*n* = 116). Means with the same letter and separately for each parameter in two successive columns (number of germinated seeds, in vivo germination (%)) are not statistically significantly (n.s.) different from each other according to Duncan’s multiple range test at *p* ≤ 0.05.

Days after Sowing	No. of Germinated Seeds after		Germination (%) after	
	Drying	Drying and Cold Storage	Drying	Drying and Cold Storage
15th	70 c	95 b	63.33 c	70 bc
30th	90 b	116 a	77.33 b	90 a
45th	90 b	116 a	77.33 b	90 a
60th	90 b	116 a	77.33 b	90 a
*p*-values (general linear model/two-way ANOVA)				
Number of sowing days (Α)	0.000 ^2^			
Seed pre-treatment (Β)	0.000 ^2^			
(Α)*(Β)	0.997 ^1^			

^1^ n.s.: *p* > 0.05; ^2^
*p* ≤ 0.001.

**Table 2 plants-09-01465-t002:** In vitro germination of *Carlina diae* seeds after 40 days of culture in MS medium with or without supplement of 750 μM GA_3_. Means with the same letter in two successive columns are not statistically significantly (n.s.) different from each other according to Duncan’s multiple range test at *p* ≤ 0.05.

	In Vitro Germination (%)	
Days of Culture	MS + GA_3_	Plain MS
4	41.38 c	10.00 d
10	96.55 a	44.00 c
20	96.55 a	70.00 b
30	96.55 a	100.00 a
40	96.55 a	100.00 a
*p*-values (general linear model/two-way ANOVA)		
Culture days in MS medium (Α)	0.000 ^1^	
GA_3_ in the medium (yes/no) (Β)	0.000 ^1^	
(Α)*(Β)	0.000 ^1^	

^1^*p* ≤ 0.001.

**Table 3 plants-09-01465-t003:** Effect of indole-3-butyric acid (IBA) concentration (0, 1000, 2000 and 4000 ppm) on rooting (%), root number/rooted cutting, root length (cm) and necrosis (%) in *Carlina diae* cuttings after 40 days under a mist propagation system. Means ± standard error (S.E.) with the same letter in a column are not statistically significantly different from each other according to Duncan’s multiple range test at *p* ≤ 0.05.

Treatments	Rooting (%)	Root Number/Rooted Cutting	Root Length (cm)	Necrosis (%)
Control	71.43 b	5.00 ± 0.58 b	4.26 ± 0.43 a	0 b
1000 ppm IBA	100.00 a	11.43 ± 1.84 ab	2.97 ± 0.24 b	0 b
2000 ppm IBA	100.00 a	16.00 ± 3.45 a	2.08 ± 0.42 b	0 b
4000 ppm IBA	57.14 c	17.00 ± 2.04 a	2.88 ± 0.51 b	42.86 a
*p*-values	0.000 ^2^	0.003 ^1^	0.010 ^1^	0.000 ^2^

^1^*p* ≤ 0.01; ^2^
*p* ≤ 0.001.

**Table 4 plants-09-01465-t004:** Seasonal variation (summer, autumn, winter) in rooting (%), root number/rooted cutting and root length (cm) of *Carlina diae* cuttings treated with 2000 ppm IBA under mist for 30 and 50 days. Means ± standard error (S.E.) with the same letter in a column are not statistically significantly different from each other according to Duncan’s multiple range test at *p* ≤ 0.05.

Season	Rooting (%)	Root Number/Rooted Cutting	Root Length (cm)	Rooting Period (Days)
Summer (June)	100.00 a	13.17 ± 1.28 a	1.82 ± 0.19 b	30 b
Autumn (November)	100.00 a	9.86 ± 1.45 b	3.01 ± 0.20 a	30 b
Winter (January)	52.38 b	9.73 ± 1.33 b	1.37 ± 0.18 c	50 a
*p*-values	0.000 ^2^	0.009 ^1^	0.000 ^2^	0.004 ^1^

^1^*p* ≤ 0.01; ^2^
*p* ≤ 0.001.

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
