# Peer review of "GIS-Facilitated Effective Propagation Protocols of the Endangered Local Endemic of Crete *Carlina diae* (Rech. f.) Meusel and A. Kástner (Asteraceae): Serving Ex Situ Conservation Needs and Its Future Sustainable Utilization as an Ornamental"

_plants, 2020, doi:10.3390/plants9111465_

Round 1

Reviewer 1 Report

An excellent study. Very well placed in a context. The authors present the results of fine-tuning propagation methods to environmental factors gathered in through GIS facilitated means for a rare endemic.

I find this paper to be quite novel and of great merit. I recommend the paper is published after these two very minor items are addressed:

Line 42: "ethnobotanical"

Line 97: please check this URL, as I could not access the site with this address. 

Author Response

Dear Reviewer,

we would like to thank you for your kind and encouraging comments. The two suggested corrections in lines 42 and 97 were made and are highlighted in the revised manuscript.

Sincerely yours,

Dr Katerina Grigoriadou

Researcher HAO-DEMETER

Institute of Plant Breeding and Genetic Resources

Box Off. 60458, Postal Code 570 01

Thermi, Thessaloniki

Greece

e-mail: kgrigoriadou@ipgrb.gr , grigokat@outlook.com 

Reviewer 2 Report

This is in fact an excellently researched and written paper that was a pleasure to read and review. 

The only minor revision suggested would be to incorporate a little more comparative literature on others studies that presented micropropagation of endangered Asteraceae, e.g. work on Argyroxiphium sandwicense or Centaurea zeybeckii.

Author Response

Dear Reviewer,

We would like to thank you for your kind and encouraging comments.

As suggested, we have incorporated bibliographic references regarding the propagation-conservation of the two species mentioned by the reviewer, i.e. Argyroxiphium virescens (Hawaian single-island endemic) and Centaurea zeybeckii (Turkish single-mountain endemic). To make contrast with the importance of our findings referring to Carlina diae (Greek single-small island endemic), we added these references in the conclusions part, so one may now read:

Extremely rare single-island or single-mountain local endemics of Asteraceae are threatened with extinction across the world [71 and 72, respectively], and in-situ conservation coupled with ex-situ effective propagation methods are the only apparent avenues to preserve them effectively [71, 72].

We hope that this addition (it is highlighted in the revised manuscript) can properly address the reviewer’s comment. Unfortunately, we were not able to find the reference of Ferguson, N. & Pavlik, B. M. 1991 [Micropropagation of Argyroxiphium virescens (Haleakala greensword) for conservation and reintroduction. I. Explant sterilization and callus initiation. Department of Biology. Oakland: Mills College] to add it.

Sincerely ours,

Dr Katerina Grigoriadou

Researcher HAO-DEMETER

Institute of Plant Breeding and Genetic Resources

Box Off. 60458, Postal Code 570 01

Thermi, Thessaloniki

Greece

e-mail: kgrigoriadou@ipgrb.gr , grigokat@outlook.com